# INTERACTIVE MODEL WITH STRUCTURAL LOSS FOR LANGUAGE-BASED ABDUCTIVE REASONING

## ABSTRACT

The abductive natural language inference task ($\alpha$NLI) is proposed to infer the most plausible explanation between the cause and the event. In the $\alpha$NLI task, two observations are given, and the most plausible hypothesis is asked to pick out from the candidates. Existing methods model the relation between each candidate hypothesis separately and penalize the inference network uniformly. In this paper, we argue that it is unnecessary to distinguish the reasoning abilities among correct hypotheses; and similarly, all wrong hypotheses contribute the same when explaining the reasons of the observations. Therefore, we propose to group instead of ranking the hypotheses and design a structural loss called "joint softmax focal loss" in this paper. Based on the observation that the hypotheses are generally semantically related, we have designed a novel interactive language model aiming at exploiting the rich interaction among competing hypotheses. We name this new model for $\alpha$NLI: Interactive Model with Structural Loss (IMSL). The experimental results show that our IMSL has achieved the highest performance on the RoBERTa-large pretrained model, with ACC and AUC results increased by about 1% and 5% respectively.

## 1 INTRODUCTION

Abductive natural language inference ($\alpha$NLI) (Bhagavatula et al. (2020)) is a newly established branch of natural language inference (NLI) and is an interesting task in the area of natural language processing (NLP) based commonsense reasoning. Originating from NLI which targets at the semantic relationship between the two sentences, $\alpha$NLI further estimates the abductive reasoning of each sentence by explicitly deducing its cause. In the past years, $\alpha$NLI has attracted increasing attentions as it makes NLP tools more explainable and comprehensible. As of today, typical applications of $\alpha$NLI include knowledge graph Completion (Yu et al. (2020)) (Bauer & Bansal (2021)), question answering (Ma et al. (2021)), sentence in-filling (Huang et al. (2020)), knowledge integration (Zhou et al. (2021)) and so on.

To better motivate this work, we have shown a comparison between NLI and $\alpha$NLI in Table 1. For NLI, the task is to judge the relationship between the premise statement P and the hypothetical sentence H based on the given information in P. Options of the answer can be implication, neutrality, or contradiction. For $\alpha$NLI, a pair of observations ($O_1$ and $O_2$) and some hypotheses (e.g., two competing hypotheses $H^1$ and $H^2$ in the example) are given. The task of $\alpha$NLI is to deduce the more plausible reason between $H^1$ and $H^2$ that can explain the situational change from $O_1$ to $O_2$. In addition to constructing the $\alpha$NLI task, the authors of (Bhagavatula et al. (2020)) has released a new challenge data set, called ART and reported comprehensive baseline performance for $\alpha$NLI by directly employing and retraining a solution for NLI, i.e., ESIM+ELMo(Chen et al. (2017), Peters et al. (2018)). They also found that the pretrained language model can apparently influence the performance of an algorithm and demonstrated some test results with the latest language models like GPT(Radford et al. (2018)) and BERT(Devlin et al. (2019)).

We note that there is still a considerable gap between the human performance and the class of baseline models in (Bhagavatula et al. (2020)). More recently, (Zhu et al. (2020)) argued that the former framework cannot measure the rationality of the hypotheses, and reformulated $\alpha$NLI as a learning-to-rank task for abductive reasoning. In their approach, RoBERTa(Liu et al. (2019)), BERT(Devlin et al. (2019)), and ESIM(Chen et al. (2017)) are all tested to work as the pretrained language model.

Table 1: Comparison of NLI tasks and $\alpha$NLI tasks, where E, N, and C represent entailment, neutral and contradiction, respectively

| Task | Context | Answer |
|------|---------|--------|
| NLI | P: A man inspects the uniform of a figure in some East Asian country.
    H: The man is sleeping. | E , N or **C** |
| | P: An older and younger man smiling.
    H: Two men are smiling and laughing at the cats playing on the floor. | E , **N** or C |
| | P: A soccer game with multiple males playing.
    H: Some men are playing a sport. | **E** , N or C |
| $\alpha$NLI | $O_1$: Dotty was being very grumpy.
    $H^1$: Dotty ate something bad.
    $H^2$: Dotty call some close friends to chat.
$O_2$: She felt much better afterwards. | $H^1$ or $\mathbf{H^2}$ |

Under this new ranking-based framework, (Paul & Frank (2020)) introduces a novel multi-head knowledge attention model which learns to focus on multiple pieces of knowledge at the same time, and is capable of refining the input representation in a recursive manner for $\alpha$NLI.

Despite the performance improvement achieved by the ranking framework, there are still some weaknesses calling for further investigation. For instance, a practical training sample (e.g., two observations and four hypotheses) from ART is shown in Figure 1. It is easy to conclude that both $H^1$ and $H^2$ are correct answers; while the other two ($H^3$, $H^4$) are false. However, in previous ranking-based $\alpha$NLI method such as $L2R^2$ (Learning to Rank for Reasoning) (Zhu et al., 2020, four hypotheses will be trained simultaneously by treating one of the two correct answers as a more correct one. Similarly, the wrong answers are also treated as a wrong one and a worse one. Meanwhile, the ranked hypotheses are trained separately, but the sum of their probabilities is set as a fixed value - e.g., the probability of correct hypothesis $H^2$ decreases when the probability of answer $H^1$ increases.

$O_1$ : Josh bought a parrot as a pet.
$O_2$ : Josh was so excited that he taught his parrot how to say it's name!

$H^1$ : Josh realized that the parrot could talk!
$H^2$ : The parrot repeated Josh's morning greeting.
$H^3$ : Josh started teaching the parrot things.
$H^4$ : He is scared of birds.

$S(H^1) > S(H^2) > S(H^3) > S(H^4)$

$L2R^2$

$S(H^1) > S(H^3) = S(H^4)$
$S(H^2) > S(H^3) = S(H^4)$

IMSL

Figure 1: Comparison of $L2R^2$ method and IMSL method. Among them, $O_1$, $O_2$ represent observations, $H^1$, $H^2$ are correct answers, $H^3$, $H^4$ are wrong answers, $S(H^i)$ represents the score of the i-th hypothesis correctness.

In this paper, we advocate a new approach for $\alpha$NLI as shown in Figure 1. Our principle of abductive reasoning is constructed based on following two arguments: 1) a hypothesis is correct because its meaning explains the change of the observations. In practice, the causes of situational changes are often diverse, and therefore the answers are seldom unique. It follows that we do not need to intentionally distinguish or rank the correct answers. 2) a hypothesis is wrong because it can not explain the cause of some event. Therefore, all wrong answers contribute the same - i.e., it is plausible to treat all wrong hypotheses equally in the process of constructing our reasoning network. We argue that the proposed abductive reasoning principle is closer to commonsense reasoning by humans than previous ranking-based approaches.

Based on the above reasoning, we propose a new abductive reasoning model called Interactive Model with Structural Loss (IMSL) for $\alpha$NLI as shown in Figure 2. The IMSL model mainly consists of two components: interactive model and structural loss. On the one hand, note that in the process

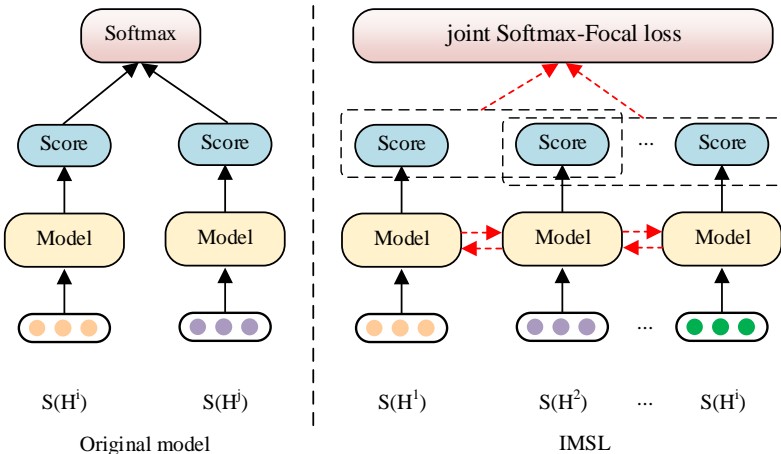

Figure 2: Comparison of the interaction between the traditional model and the IMSL model, where S (H$^i$) represents the input sequence containing the i-th hypothesis

of extracting the language features of an arbitrary hypothesis, its relationship to other hypotheses should also be considered because the hypotheses are often semantically related Pearl (1986). To this end, we can design an information interaction layer to capture the relationship between different hypotheses and produce more discriminative language feature vectors. On the other hand, we have constructed a new loss function called "joint softmax focal loss" inspired by a recent work (Lin et al. (2017)). It is essentially a structural softmax based Focal loss formed by sequentially constructing a loss for each score group that composed by a correct hypothesis and all wrong hypotheses. When compared with conventional models, we argue that IMSL is more powerful for the task of $\alpha$NLI by jointly exploiting the rich relation among competing hypotheses. The main technical contributions of this work can be summarized as follows.

1) For $\alpha$NLI task, we claim that, the correct hypotheses of a given observation pair are often diverse, and there is no need to tell them apart. The wrong hypotheses contribute the same to the task. We regroup instead of ranking all hypotheses, as shown in Figure 1.

2) Aiming at the problem of incorrect probability distribution between correct hypotheses in the training process, a joint softmax focal loss is proposed. For the hypotheses groups formed in the rearrange process, we design a softmax-based focal loss for each group and combine them into a joint loss.

3) In view of the problem that traditional models cannot capture the language relationship between different hypotheses, we have added an information interaction layer between different hypothesis models. The information interaction layer increases the area under the receiver's operating characteristic curve (AUC) by about 5%.

4) Impressive abductive reasoning performance is achieved by IMSL when tested using RoBERTa as the pretrained language model. The best language model DeBERTa (He et al. (2021)) is not tested due to the constraint by our limited GPU resources (4-piece RXT 2080Ti). In our experiment, compared with all recent algorithms whose codes have been made publicly available, the IMSL method has achieved state-of-the-art results in ACC and AUC on both the validation set and test set. Besides, on the public leaderboard[1], IMSL is the best non-DeBERTa based algorithm and ranks 4/56 in all (including both DeBERTa based and non-DeBERTa based) competing methods.

## 2 RELATED WORK

$\alpha$NLI task solves an abductive reasoning problem based on natural language inference (NLI). In the past years, there has been an explosion of NLI benchmarks, since the Recognizing Textual Entail-

---

[1]https://leaderboard.allenai.org/anli/submissions/public

ment (RTE) Challenges was introduced by (Dagan et al. (2005)) in the early 2000s. Then, in order to find the most reasonable explanation of the incomplete observations, (Bhagavatula et al. (2020)) studied the feasibility of language-based abductive reasoning and proposed the task of $\alpha$NLI. It pays more attention to the information provided in the premise than the RTE task. For traditional RTE, the main task is to judge the relationship between the premise sentence and the hypothetical sentence, but the main objective of $\alpha$NLI is to select the most plausible hypothesis among the hypotheses given two observations.

$\alpha$NLI is the first language-based abductive reasoning study. This shift from logic-based to language-based reasoning draws inspirations from a significant body of works on language-based entailment (Bowman et al. (2015); Williams et al. (2018)), language-based logic (Lakoff (1970); MacCartney & Manning (2007)), and language-based commonsense reasoning (Mostafazadeh et al. (2016); Zellers et al. (2018)). In addition to establish $\alpha$NLI, (Dagan et al. (2005)) have also released a new challenge dataset, i.e., ART, which can be visited through the first footnote in this paper. The authors have also formulate the task as a multiple-choice task to support easy and reliable automatic evaluation. Specifically, from a given context, the task is to choose the more reliable explanation from a given pair of hypotheses choices.

However, discriminating correct from wrong does not measure the plausibility of a hypothesis in $\alpha$NLI (Zhu et al. (2020)). So, to fully model the plausibility of the hypotheses, Zhu et al. turn to the perspective of ranking and propose a novel learning to rank for reasoning (L2R$^2$) approach for the task. The authors rank the hypotheses based on the number of times they appear in the dataset, and use some pairwise rankings as well as a listwise ranking as loss. Pairwise rankings contains Ranking SVM (Herbrich et al. (2000)), RankNet(Burges et al. (2005)), LambdaRank(Burges et al. (2006)), and Listwise Ranking contains ListNet(Cao et al. (2007)), ListMLE (Li et al. (2020)) and ApproxNDCG(Qin et al. (2010)). The experiments on the ART dataset show that reformulating the $\alpha$NLI task as ranking task really brings obvious improvements. After that, (Paul & Frank (2020)) proposes a novel multi-head knowledge attention model that encodes semi-structured commonsense inference rules and learns to incorporate them in a transformer based reasoning cell. The authors still prove that a model using counterfactual reasoning is useful for predicting abductive reasoning tasks. Accordingly, they have established a new task called Counterfactual Invariance Prediction (CIP) and provide a new dataset for this.

In addition to the abductive reasoning models, the pre-trained language model still plays an important role in $\alpha$NLI task. Early ways for language inference are constructed directly by some simple statistical measures like bag-of-words and word matching. Later, various kinds of neural network architectures are used to discover useful features in the languages, like word2vec(Mikolov et al. (2013)) and GloVe(Pennington et al. (2014)). Recent works have developed contextual word representation models, e.g.,Embeddings from Language Models (ELMO) by Peters et al. (2018) and Bidirectional Encoder Representations from Transformers(BERT) by Devlin et al. (2019). The original implementation and architecture of BERT has been outperformed by several variants and other transformer-based models, such as RoBERTa, DeBERTa and UNIMO. RoBERTa(Liu et al. (2019)) replaces training method of BERT and uses larger batches and more data for training. DeBERTa(He et al. (2021)) uses the disentangled attention mechanism and an enhanced mask decoder to improves the BERT and RoBERTa models. In order to effectively adapt to unimodal and multimodal understanding task, Li et al. (2021) proposes the UNIMO model. In this paper, however, restricted by our computing resources, RoBERTa is selected as our language model.

## 3   INTERACTIVE MODEL WITH STRUCTURAL LOSS (IMSL) METHOD

IMSL model consists of two components: context coding layer and information interaction layer (the backbone network) as well as a joint softmax focal loss (objective function). The design of the model architecture and loss function are described in detail below.

### 3.1   INFORMATION INTERACTION MODEL

**Model input:** Under the framework of IMSL, a training sample $X$ includes two given observations (i.e., $O_1$ and $O_2$) and a group of candidate hypotheses denoted by $\mathbf{H} = \left\{ H^j \right\}_{j=1}^{N}$ ($N$ is the number of candidate hypotheses). Then, binary labels $\mathbf{y} = \{y_j\}_{j=1}^{N}$ are assigned to each hypoth-

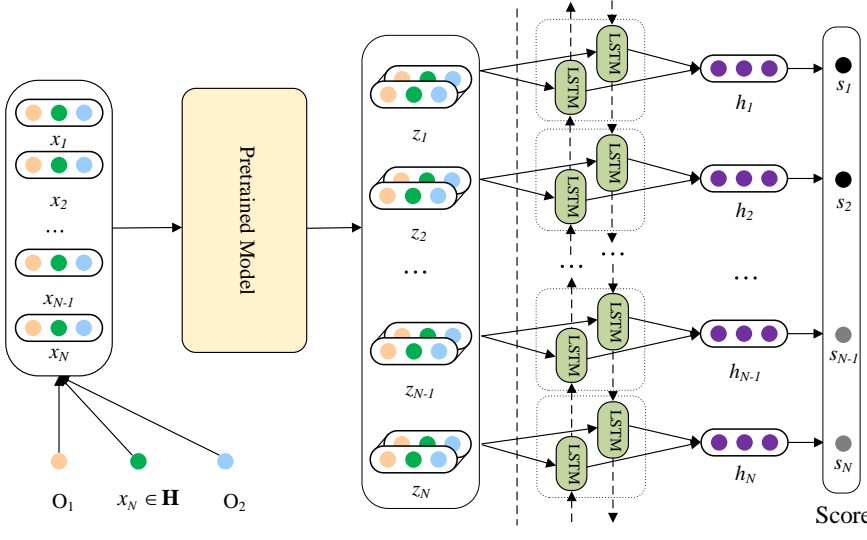

Figure 3: The proposed IMSL model consists of a context coding layer (using a pre-trained model) and an information interaction layer (characterizing the relationship among different hypotheses).

esis by $y_j = 1$ when $H^j$ is the correct, and $y_j = 0$ when $H^j$ is the wrong. The task of abductive reasoning can be characterized by a mapping from $X$ to $\mathbf{y}$. For explicitly estimating the relation between each hypothesis and the two observations, we can construct a triad for each hypothesis as $x_j = \left[ O_1; H^j; O_2 \right] \left( H^j \in \mathbf{H} \right)$. This way, each sample in the training set $X$ can be represented by $[x_1, x_2, \cdots, x_N] \rightarrow [y_1, y_2, \cdots, y_N]$.

**Context coding layer:** We use a pre-trained language model (RoBERTa-large is used in our experiment) to calculate the contextual representation of the text. For each word in a single input $x_j$, an embedding vector with context information is generated. For each sentence in a single input $x_j$, a sentence-level embedding matrix $v_j = \text{encode}(x_j)$ is first obtained, where $\text{encode}(\cdot)$ denotes the pre-trained model for encoding. Then we can sum the word embedding dimensions in the feature matrix to generate the feature vector $z_j$.

**Information interaction layer:** Traditional models only consider one single input $x_j$ during scoring as shown in Fig. 2, which makes it difficult to capture the relationship between different inputs (e.g., $x_j$ and $x_{j+1}$). To exploit the dependency between two different inputs, we propose to construct a novel information interaction layer as follows. First, a pair of feature vectors $z_j$ and $z_{j+1}$ can be generated after $x_j$ and $x_{j+1}$ are passed through the context encoding layer. Second, we plug $z_j$ and $z_{j+1}$ into the information interaction layer and use BiLSTM to acquire the distributed feature representation $f_j$ and $f_{j+1}$. Finally, a fully connected module outputs the corresponding scores $s_j$ and $s_{j+1}$. A flowchart of the context coding and information interaction layers is shown in Figure 3.

To efficiently use contextual information, we use $z_j$ as the input of BiLSTM, which aim at exploiting the dependency relationship between the feature vectors. BiLSTM uses a forward LSTM and a backward LSTM for each sequence to obtain two separate hidden states: $\overrightarrow{h_j}, \overleftarrow{h}_j$. The key idea of BiLSTM is to generate the final output at time $t$ by concatenating these two hidden states:

$$h_j = \left[ \overrightarrow{h_j}, \overleftarrow{h}_j \right]. \tag{1}$$

After passing the BiLSTM layer, we can get a bidirectional hidden state vector $h_j$, then use the fully connected layer to generate the final score $s_j$. For computational efficiency, a linear regression formula is adopted here for prediction score:

$$s_j = W_j \cdot h_j + b_j, \tag{2}$$

where $W_j \in \mathbb{R}^{2d \times d}, b_j \in \mathbb{R}^d$.

### 3.2 JOINT SOFTMAX FOCAL LOSS FUNCTION

Based on the output score layer of the IMSL model, we propose to design a new structural loss function based on the principle of abductive reasoning. Instead of ranking-based approach, the proposed loss function for each sample is formulated as a linear combination of softmax focal losses for several rearranged groups, which is called joint softmax focal loss. The detailed procedure of generating multiple rearranged groups is shown in Figure 4. We note that it is unnecessary to compare the group of correct hypotheses; while the exploration of the relationship between correct hypothesis and wrong hypotheses is sufficient for the task of score prediction. Therefore, we can rearrange the set of $N$ prediction scores into several groups, each of which only contains a single correct hypothesis. A toy example is given in Figure 4 where the two hypotheses are correct, and all other hypotheses are wrong. In this example, the total $N$ scores can be divided into two groups associated with two correct hypotheses, respectively.

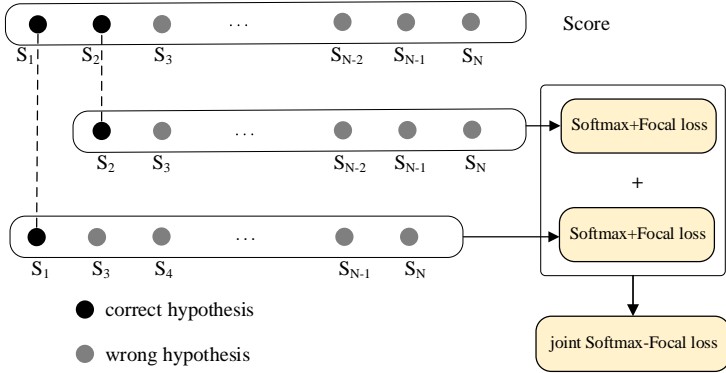

Figure 4: An example of rearrange groups for joint softmax focal loss.

With the above construction, we can first apply the softmax operation to each rearranged group and then combine them into a joint loss function. In the first step, each prediction score is given by:

$$
\hat{y}_n = \begin{cases} \frac{e^{s_n^1}}{e^{s_n^1}+\sum_i e^{s_i^0}}, & if \quad y_n = 1. \\ \sum_j \frac{e^{s_n^0}}{K\left(e^{s_j^1}+\sum_i e^{s_i^0}\right)}, & if \quad y_n = 0. \end{cases}
\tag{3}
$$

where $y_n$ is the correct/wrong label, and $\hat{y}_n$ represents a predicted value. The normalization factor $K = \sum_j y_j$ represents the number of correct hypotheses. Note that $s_i^0$ represents the scores of the wrong hypotheses, where $i$ is the position of the false label. Similarly, $s_i^1$ indicates the score of the correct hypotheses.

In addition to the softmax loss, we have borrowed the idea of focal loss from (Lin et al. (2017)) and introduce a balancing factor $a \in (0, 1)$ to control the shared weight of the correct hypothesis and the wrong ones. Here, $a$ is used for the correct hypotheses, and $1 - a$ is used for the wrong hypotheses, i.e.,

$$
\beta_n = y_n \cdot a + (1 - y_n)(1 - a).
\tag{4}
$$

Putting things together, we can rewrite the joint softmax focal loss as

$$
\mathcal{L} = F_l(y, \hat{y}) = -\sum_n \beta_n \cdot (1 - p_n)^\gamma \cdot \log(p_n).
\tag{5}
$$

where

$$
p_n = y_n \cdot \hat{y}_n + (1 - y_n)(1 - \hat{y}_n) + \varepsilon.
\tag{6}
$$

Here, the parameter $\gamma$ is included for regulating the model's attention to hard hypotheses during the training of IMSL model. As suggested in (Lin et al. (2017)), $\gamma \in [0.5, 5]$. Then, a small positive real number $\varepsilon$ of $1e$-8 is used to avoid the numerical instability problem. In practice, both $a$ and $\gamma$ are used as hyper-parameters which can be tuned by cross-validation. For the example shown in Figure

4, by assuming that the softmax focal losses for the two groups are $\mathcal{L}_{group1}$ and $\mathcal{L}_{group2}$, we can obtain the overall loss by $\mathcal{L}_{sample} = \mathcal{L}_{group1} + \mathcal{L}_{group2}$. Furthermore, the total loss for all training samples can be estimated by the sum of the losses over individual samples.

## 4 EXPERIMENT

In this section, the experimental results on public data sets are presented to evaluate the method proposed in this paper.

In recent years, more and more Pre-Training models have been proposed, such as DeBERTa (He et al. (2021)), UNIMO (Li et al. (2021)), etc. They use more data for training and have more parameters. Due to limited computational resources, we did not conduct comparative experiments with these high-performing yet computationally demanding pretrained models.

**Evaluation indicators:** AUC and ACC are the most common evaluation indicators. Since the original ACC cannot evaluate the model that is far away from the test data, AUC is added as an additional evaluation index to handle skewed sample distribution. AUC is a measurement method that is statistically consistent and more discriminative than ACC.

### 4.1 EXPERIMENTAL SETUP

Tasks and settings: The $\alpha$NLI task uses the ART dataset, which is the first large-scale benchmark dataset used for abductive reasoning in narrative texts. It consists of about 20,000 observations and about 200,000 pairs of hypotheses. The observations come from a collection of manually curated stories, and the hypotheses are collected through crowdsourcing. In addition, the candidate hypotheses for each narrative context in the test set are selected through the adversarial filtering algorithm with BERT-L (Large) as the opponent. The input and output formats are shown in Table 2.

Table 2: The format of data input and output in $\alpha$NLI task

| Task | Input Format | Output Format |
|------|-------------|---------------|
| $\alpha$NLI | [CLS] $O_1$ [SEP] $H^i$ [SEP] $O_2$ [SEP] | $H^1$ or $H^2$ |

**Hyperparameter details:** Due to the difference in the amount of data, the focusing parameter and the amount of training data will vary. For different training data, select the hyperparameter that produces the best performance on the test set. Specifically, the learning rate is fixed at 1e-6, the batch size is fixed at 1, and the training batch will vary with the amount of training data. Training uses Cross Softmax+Focal Loss. For the validation set, ACC and AUC are used for evaluation. Use the results of five different seeds to evaluate the performance of the test set.

**Baseline:** We have used the following four baseline models for comparison: A) BERT (Devlin et al., 2019) is a language model that uses a masked language model and predicts the next sentence as the target training. For example, it masks certain words in the input, and then trains it and predicts the words that are blocked. B) RoBERTa (Liu et al., 2019) has the same structure as BERT, but there is no prediction (NSP) for the next sentence. RoBERTa-B (ase) and RoBERTa-L (arge) use more data and larger batches for training. C) Learning to Rank for Reasoning (L2R$^2$) Zhu et al. (2020) reprogrammed the $\alpha$NLI task as a ranking problem, using a learning ranking framework that includes a score function and a loss function. D) Multi-Head Knowledge Attention (MHKA) (Paul & Frank, 2020) proposed a new multihead knowledge attention model, and used a novel knowledge integration technology.

### 4.2 EXPERIMENTAL RESULTS

Our experimental results in the $\alpha$NLI task are shown in Table 3. The baseline comparison models are: Majority, GPT, BERT-L, RoBERTa-L, L2R$^2$ and MHKA related results. It can be observed that the IMSL method improves about 3.5% in ACC and about 7.5% in AUC compared with RoBERTa-L. The results show that the improvement of ACC is mainly attributed to the new IMSL loss function,

Table 3: Results on the $\alpha$NLI task: The results are quoted from (Bhagavatula et al., 2020), L=Large

| Model | Dev(ACC%) | Dev(AUC%) | Test(ACC%) |
|---|---|---|---|
| Human Perf | - | **-** | 91.40 |
| Majority | 50.80 | - | - |
| GPT | 62.70 | - | 62.30 |
| BERT-L | 69.10 | 69.03 | 68.90 |
| RoBERTa-L | 85.76 | 85.02 | 84.48 |
| L2R$^2$ | 88.44 | 87.53 | 86.81 |
| MHKA | 87.85 | - | 87.34 |
| Ours | | | |
| RoBERTa-L+IMSL | **89.20** | **92.50** | **87.83** |

and the improvement of AUC is mainly attributed to the exploitation of the relationship between the hypotheses by the proposed information interaction layer.

**Low-resource setting**: Testing the robustness of the model to sparse data on $\alpha$NLI tasks refers to the low-resource scenario where the MHKA model uses $\{1,2,5,10,100\}$% training data respectively. Figure 5 shows how the model improves on MHKA, RoBERTa-Large, and L2R$^2$. The experimental results show that the model in this paper can achieve better results in the case of low resource setting. When using 1% training data only, the improvement brought by IMSL is the most significant, which is about 4% higher than that of L2R$^2$ and MHKA. Experimental results show that our method performs consistently better than other competing methods on low-resource data sets.

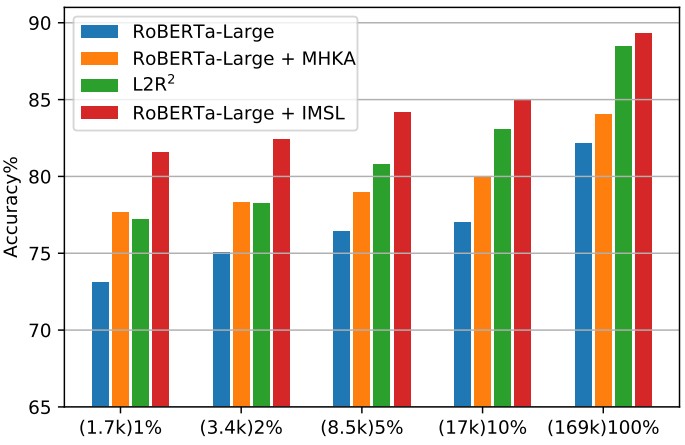

Figure 5: The accuracy of $\alpha$NLI under low resource settings

## 5 DETAILED ANALYSIS VIA ABLATION STUDY

To more clearly show the contribution of each module, we have done corresponding comparative experiments on both information interaction layer and hyperparameter tuning.

### 5.1 ABLATION STUDY OF THE INFORMATION INTERACTION LAYER

First, we have conducted ablation study experiments on the related BiLSTM to investigate the role played by the information interaction layer. The hyperparameters of Focal Loss will be fixed to reduce the impact on BiLSTM. Through the experimental results, it can be found that the addition of BiLSTM greatly improves the AUC, but does not have a significant impact on ACC. The following

Figure 6 shows the visualization results on the validation set. In the plot, the abscissa is the score of hypothesis 1 and the ordinate is the score of hypothesis 2. The red points in the upper left corner correspond to the subset of correct hypotheses, so do the blue points in the lower right corner. It can be seen that the introduction of the information interaction layer pushes all points further away from each other and toward the four corners. It follows that the margin between the positive and negative samples is larger, implying improved discriminative power.

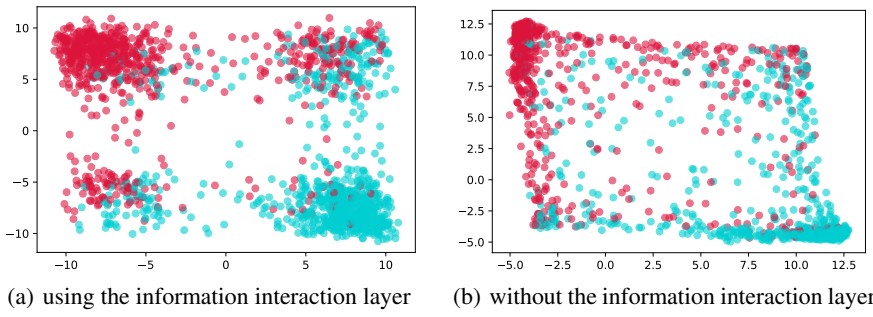

(a) using the information interaction layer      (b) without the information interaction layer

Figure 6: Score distribution using and without using the information interaction layer

## 5.2 PARAMETER COMPARISON

When we select the parameters of Focal Loss, several experiments were carried out on the two hyperparameters of the balancing factor and the focusing parameter. The focusing parameter $\gamma$ in Eq. equation 5 can automatically down-weight the contribution of easy examples during training and rapidly focus the model on hard examples; while the balancing factor $0 < a < 1$ in Eq. equation 4 controls the tradeoff between correct and incorrect hypotheses. Figure 7 below shows the ACC performance of IMSL model with different focusing parameters and balance factors. In this study, $\{1, 2, 3\}$ is used as the option of focusing parameter, and $\{0.45, 0.5, 0.55\}$ is used as the set of balancing factor. It can be observed that as the most effective parameter couple is given by $\gamma = 2$, $a = 0.55$.

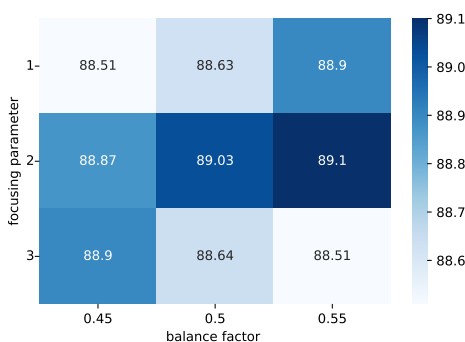

Figure 7: The ACC results of adjusting the balance factor $\gamma$ and focusing parameter $a$.

## 6 SUMMARY

In this paper, an IMSL method is proposed for commonsense abductive reasoning. It includes an information interaction layer that captures the relationship between different hypotheses, and a joint loss for our proposed way of grouping the correct/wrong hypotheses. Experimental results show that on $\alpha$NLI tasks, IMSL has better performance on ACC and AUC, especially in low-resource settings, IMSL can significantly improve the accuracy.

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
