# OpenReview forum: "Interactive Model with Structural Loss for Language-based Abductive Reasoning"
_ICLR.cc/2022/Conference — ICLR 2022 Submitted_

### Official Review · Reviewer_W1Sp · 2021-11-01

**Correctness:** 2
**Technical Novelty And Significance:** 1
**Empirical Novelty And Significance:** 2
**Recommendation:** 3
**Confidence:** 3

**Main Review:**

Strengths:

- The authors propose to use common techniques for modeling the abductive NLI task, which might be potentially useful for improving the performance of pre-trained language models.

Weaknesses:

- The writing is not clear and the paper has a lot of grammatical errors and stylistic issues in particular around citations. It needs proofreading.
- It seems that the proposed approach lacks novelty where the authors use bi-directional modeling and focal loss that are common techniques in the field. Even if the improvement on the task is significant, its contribution might be incremental.
- Given a small improvement (1% for the test accuracy) against the previous best model (MHKA), the standard deviation should be reported.
- In Section 5.1, the authors mention that ablating the component of modeling hypotheses together doesn't have a significant impact on the test ACC. Given that the model should be evaluated on the test set rather than the dev set, the authors could not conclude that the proposed approach significantly improves the task performance.
- There is no qualitative analysis on examples where modeling hypotheses together is expected to be meaningful and better than re-ranking them.

Comment:
- Use \cite and \citet properly.

**Summary Of The Paper:**

This paper proposes a method for the abductive natural language inference task. Different from previous state-of-the-art models that score and rank candidate hypotheses, the authors propose to model them together and compute a joint loss. The proposed method implemented on top of RoBERTa-large shows improvement by 1% accuracy in the test set from the previous best method, and 3% improvement from the simple RoBERTa-large.

**Summary Of The Review:**

Lack of novelty, not a significant improvement, and no qualitative analysis on the authors' hypothesis (i.e., it's useful to model hypotheses together).

---

### Official Review · Reviewer_3Vwo · 2021-11-02

**Correctness:** 2
**Technical Novelty And Significance:** 2
**Empirical Novelty And Significance:** 2
**Recommendation:** 3
**Confidence:** 4

**Main Review:**

Strengths:
The paper proposes a state-of-the-art model on the ART dataset which ranks the fourth in the leaderboard (see footnote 1 in the paper), only lower than UNIMO and DeBERTa based models.
Weaknesses:
(1) The proposed joint softmax focal loss seems not applicable to the experimental ART dataset. If I do not misunderstand, the footnote 1 provides a link to the leaderboard of the ART dataset. However, the experimental dataset in this leaderboard consists of only instances having exactly two hypotheses, where one is correct and the other is wrong. The joint softmax focal loss is designed specifically for instances having three or more hypotheses where the correct hypotheses are multiple. In other words, this proposed loss is not evaluated in the experiments.
(2) The ablation study for the BiLSTM-based information interaction layer is unclear. The main results are shown in Figure 6, but it is unclear what the red points are and what the blue points are. It is said in the paper that both the red points and blue points refer to correct hypotheses. It is also unclear why the separation of major red points and major blue points into two different corners implies a better performance.
(3) The evaluation on the effectiveness of the information interaction layer is also incomplete. It is insufficient to show this effectiveness based on a single pretrained language model RoBERTa-Large. Although the authors declare that they have no experimental resources for UNIMO and DeBERTa, they could still conduct experiments on UNIMO-base and DeBERTa-base to show that extending these base models (requiring similar computation resources as RoBERTa-Large) outperforms the corresponding baselines.


**Summary Of The Paper:**

This paper proposes a new model for the abductive natural language inference (alphaNLI) task. The model extends a pretrained language model with a BiLSTM-based information interaction layer and the joint softmax focal loss. Experimental results on the ART dataset show that the proposed model built on top of RoBERTa-Large outperforms its baseline RoBERTa-Large as well as state-of-the-art learning-to-rank-based models.

**Summary Of The Review:**

Please provide a short summary justifying your recommendation of the paper.
The paper has marginal novelty on two extensions of a pretrained language model, but one extension (the BiLSTM-based information interaction layer) is not completely evaluated, while the other (the joint softmax focal loss) is even inapplicable to the experimental dataset. This makes the contributions of the paper unconvincing at the current stage.

---

### Official Review · Reviewer_imYG · 2021-11-02

**Correctness:** 3
**Technical Novelty And Significance:** 2
**Empirical Novelty And Significance:** 2
**Recommendation:** 3
**Confidence:** 4

**Main Review:**

Strengths:
- The approach shows significant improvements on ACC/AUC compared to previous methods.
- The approach seems to be motivated well. It is reasonable that it might not be helpful to distinguish amongst the set of correct hypotheses and the set of incorrect hypotheses.

Weaknesss:
- Presentation: The paper is not written in a clear and concise manner. For example, there are some loose claims about abductive natural language inference in the first paragraph. Description of previous work such as Paul and Frank, 2020 is not presented clearly. Finally, there are some details that might not be necessary to include in the paper (description of BiLSTM, repetitions in description of alphaNLI). Also, Figure 2 is not easy to understand and I would suggest revising it.
- Impact: Do the authors think their approach could be used for tasks other than alphaNLI? I'm a bit concerned that the proposed approach is too specific to the task of alphaNLI. While a better approach specific for alphaNLI is valuable in itself, but it does limit the generality of the approach.
- Results: Some of the results are missing. The AUC/ACC scores corresponding to ablation in sec 5.1 are missing. Also, the experiment in sec 5.2 is not particularly useful and could simply be summarized by saying that the "approach is not as sensitive to hyperparameter settings".
- Error analysis: It seems like model performances on the ART task are nearing human performance. In this scenario, it would have been helpful to see some examples where the model still fails to predict the right answer and also examples where their proposed model helps. There is lack of any error analysis presented in the paper.

Questions:
- Can you provide the AUC/ACC scores corresponding to ablation performed in sec 5.1?
- Do you have any thoughts on how one might adapt this framework for abductive natural language generation?

Other typos / language:
- For in-line citations, please use \citet.

**Summary Of The Paper:**

This paper proposes a novel method called Interactive Model with Structural Loss (IMSL) to tackle the task of abductive natural language inference. Their approach is based on the observation that it may be unnecessary to intentionally distinguish among the set of correct hypotheses and the set of incorrect hypotheses. Specifically, they propose a novel loss function called the joint softmax focal loss function (based on Lin et al, 2017) that groups together a single correct hypothesis together with all the incorrect hypotheses. Their approach first uses a pretrained language model to get encodings for each of the hypotheses which are then passed to a BiLSTM to get scores for each hypothesis. The first component is called the context coding layer and the second one is called an information interaction layer.

The authors perform experiments on the ART dataset from Bhagavatula et al, 2020 and show improvements on area under curve (AUC) and accuracy (ACC) compared to previous methods. Further, they present an ablation experiment where they remove the information interaction layer and show that it helps with distinguishing correct and incorrect hypotheses.

**Summary Of The Review:**

I think the paper proposes a useful method for the task of abductive natural language inference based on good intuition.
However, I'm afraid the impact of this work might be limited (disregarding the fact that larger models like DeBERTa outperform the proposed approach) and the presentation/analyses needs quite a bit of work.

---

### Official Review · Reviewer_hk5y · 2021-11-03

**Correctness:** 2
**Technical Novelty And Significance:** 3
**Empirical Novelty And Significance:** 2
**Recommendation:** 3
**Confidence:** 3

**Main Review:**

*Strengths*

S1) The paper obtains a high score on the $\alpha$NLI dataset, surpassing most models on the public leaderboard except for some recent ones that use improved pre-trained representations (DeBERTa and UNIMO).

*Weaknesses*

W1) I have one major question about the evaluation (see question Q1a below) which makes me wonder whether the results are comparable to some of the past work (e.g. the leaderboard results cited).

W2) It was really difficult to tell what's responsible for the improvement in results over past work, especially given that I found it difficult to understand the motivation for parts of the approach (see below). I would have really appreciated some ablation experiments that individually removed the parts (the info interaction layer, and the focal loss) individually and in combination to help explain what's the cause of these strong results (and give more confidence that these strong results on this single dataset aren't just due to e.g. better hyperparameter tuning or optimization). Section 5.1 makes it seem like one of these ablations was done, but there don't seem to be any evaluation results reported from it which struck me as odd.  The closest the paper came to presenting evaluation results was the RoBERTa baseline in the results table, but the paper didn't give any details (that I could find) about this RoBERTa baseline's implementation in comparison to the full proposed method.

W3) The model was quite complex, and a few choices seemed strange to me.
- Is the ordering of the candidate hypotheses 1...N significant? If so, how is it determined? If not, why use a BiLSTM in the information interaction layer?
- There's a number of changes from the L2R^2 method, if I understand correctly (at least from the way L2R^2 was described): switching from a total ordering to a partial ordering, switching from a ranking loss to a softmax, and using a focal loss. While I feel like the introduction and Figure 1 help motivate using a partial ordering, I wasn't clear about why the other changes are necessary, or if they help (e.g. why not use one of the more standard pairwise rankings cited in Section 2)?
- Why use a focal loss? If I understand correctly, focal loss is motivated in class imbalance, but it wasn't totally clear to me where that imbalance comes from here.

W4) I felt like the impact of the paper is likely to be limited, with the paper in its current form: it's a relatively complex model for one single task, with little analysis of why the method improves or indication that the approach could be more broadly useful, and it relies heavily on one feature of this dataset (having multiple annotated hypotheses for a single pair of observations).

W5) Parts of the writing were pretty unclear to me. In particular I found it difficult to follow the description of the joint softmax layer in section 3.2 (see questions below), and (to a lesser extent) the information interaction layer.

*Questions*

Q1) How are predictions made from the model during evaluation? In particular:

Q1a) Is the interaction layer used in evaluation, and if it is used, what candidate hypotheses are used for a given observation pair? If I understand the original $\alpha NLI$ setup, there are just a pair of hypotheses for each obesrvation pair (a binary classification task); is that also the setup here in evaluation (I understand it's not, in training)?

Q1b) How is the classification obtained in evaluation? Which quantity in section 3.2 is used to select a hypothesis?

Q2) In Section 4.1,  "For different training data, select the hyperparameter that produces the best performance on the *test set*". I'm assuming this is a typo, but just to confirm, what data split was used to tune the hyperparameters (validation or testing)?

Q3) Section 4.1: "Use the results of five different seeds to evaluate the performance of the test set". What does this mean -- how were the seeds used? To compute an average? If so, it would help to report standard deviations along with the results in Table 3.

Q4) "Specifically, the learning rate is fixed at 1e-6, the batch size is fixed at 1, and the training batch will vary with the amount of training data." What does it mean to have a varied training batch but a fixed batch size? And does "amount of training data" here refer to the low-resource setting at the end of 4.2, or something else?

Q5) I was pretty confused by the $s_i^0$ and $s_i^1$ notation in section 3.2, and in general was unclear on how Equation (3) was motivated or derived. It would be really helpful to explain this in more detail.

Q6) Could more details be given about training: what optimizer is used, and is any early stopping performed (and if so what's the stopping criterion)?


**Summary Of The Paper:**

This paper proposes a model and loss function for the task of abductive natural language inference ($\alpha$NLI), which is a commonsense reasoning task that requires a model to choose which of two possible *hypothesis* sentences best goes in between two *observation* sentences, to explain the second observation (e.g. for observation sentences "Josh bought a parrot" and "Josh taught his parrot how to say its name", selecting the hypothesis "Josh realized the parrot could talk" over "He is scared of birds").

The paper, following past work (Zhu et al. 2020), uses the fact that the training set for this dataset contains multiple instances with the same pair of observations (but different hypotheses), allowing training using a ranking loss rather than just a binary classification loss. This paper proposes to use a BiLSTM to aggregate information across RoBERTa encodings of hypotheses for the same observations, groups together subsets of the hypotheses, and applies the focal loss of Lin et al. 2017 on top of a grouped softmax.

The full model improves performance substantially on the dataset in comparison to baselines and other past work that also uses a RoBERTa encoder.

**Summary Of The Review:**

I didn't feel like this paper is strong enough in its current form -- while it obtains strong results on the $\alpha$NLI dataset, I felt like there's unfortunately not much here that other work would be able to build on. I had trouble understanding why the proposed method does well, or whether it would be likely to work on other tasks (or even other datasets for the same task if they lacked the multi-view property this one has). The paper would be a lot stronger with clearer motivation for the technique, some ablations and more thorough comparisons to baselines, and improved writing.

---

### Official Review · Reviewer_cRzc · 2021-11-06

**Correctness:** 3
**Technical Novelty And Significance:** 2
**Empirical Novelty And Significance:** 2
**Recommendation:** 3
**Confidence:** 4

**Main Review:**

### Strengths of the paper
- The paper is overall well written. The text describes the problem well. The proposed model changes are easy to follow as well as the focal loss.

- The experimental analysis show strong, but not state-of-the-art results, on the aNLI (ART) dataset.

- The most interesting result, to me, are the results on the artificially low resource setting (1-2% of the training data). This result shows there seems to be some value of the proposed approach in improving sample efficiency during learning.

### Weaknesses of the paper

- One main point that I struggled to understand from this work is whether the techniques proposed here are specific to the ART dataset, or whether they make sense to apply in other settings. Are there other multiple-choice tasks in which the combination of late-stage interaction of representations and focal losses improves learning over, for example, learning-to-rank approaches?

- Another primary concern relates to the modest overall improvement in performance for ART. Specifically, it seems that ART, by itself, is not such a useful task. But it is a useful task to (1) evaluate progress in natural language understanding more generally, and (2) may transfer learning to other tasks. In fact, in the original aNLI paper (https://arxiv.org/abs/1908.05739 ) Section 7 describes how learning from ART can be transferred over to other tasks. This work does not show whether the new techniques will build better representations to transfer to new tasks or whether better performance on ART is derived solely from new information learned in the newly introduced interaction layers.

- I found Section 5 a bit lacking in terms of details that I would have expected in order to fully comprehend the contributions of this work. For example, Section 5.1 attempts to study the contribution of the interaction layer. However, besides the scatter plot in Figure 6 (which is quite hard to interpret, it doesn’t seem convincing that (a) discriminates items better than (b)), we don’t have a quantitative analysis of the impact of the interaction layer. How much of the performance is degraded/improved by adding or removing the interaction layers? This crucial information is missing,

- Similarly in Section 5.2, we see the results of modifying some hyperparameters for the focal loss. However, the most important result in my opinion, is observing the effect of removing the focal loss altogether (falling back to simpler cross-entropy softmax or learning-to-rank loss).
- As it stands, it is not possible to infer how much of the improvement described in Section 4.2 (Table 3 and Figure 5) is derived from the interaction layer and how much is derived from the focal loss.

- One final weakness of this work is that none of the two techniques used here (interaction layers and focal loss) are new to this paper, as the Related Work section of this paper clearly states. What is new is their application to the aNLI task. As a result I would have expected a much more thorough analysis of their impact on this task, clearly outlining why these techniques should be applied in tandem when modeling abductive reasoning.

### Other comments

- In Section 3, the per-example representation is derived in the “context coding layer” by summing the individual word (token?) representation. Is this correct? If so this seems quite unintuitive, and warrants more explanation. In particular when pooling representations, particularly of variable-length sequences, there is a normalization problem that occurs (e.g., short vs. long sentences contain different numbers of tokens). That is why pooling methods use average or max, or attention, or even a reserved [CLS] token that can attend to the entire sequence. I would have liked to see some explanation for this choice, and why this method is expected to work well.

- It would have been useful to compare how the focal loss compares to a simpler weighted binary cross-entropy loss where each positive is compared to each negative in the batch.

- To improve the paper, I would have focused more on the low resource setting which seems to be the main clear benefit of the proposed techniques (Figure 5). In my opinion, the impact of this work could be much greater by (1) finding other related tasks where a  multiple-choice task is being modeled, and (2) comprehending whether they also improve in low resource settings with the techniques described here. With better analysis of the impact of late-stage interaction layer vs. focal loss, one may be able to get more insight into why sample efficiency seems to be improved. Finally, I would focus on a qualitative analysis of the wins in the low resource setting attempting to classify the types of examples that are improved using the techniques here versus a L2R2 or other approaches.


**Summary Of The Paper:**

This paper proposes a new model architecture, along with a new loss, to solve the abductive natural language inference (aNLI) task. Specifically, this work hypothesizes that models benefit from the interaction between multiple choice options, and thus introducing a “information interaction layer” to allow the interaction of representations of individual options. Further, it proposes the use of a “focal loss” to rebalance the loss weight between positives and multiple negative examples during training. This work evaluates proposed changes on the ART dataset, extending the evaluation to an artificial low-resource setting. Finally, the paper presents preliminary analysis of the effects of the interaction layer and focal loss on the results.


**Summary Of The Review:**

While this study has some preliminary results that indicate that late-stage interaction between options in a multiple-choice task setting is useful, I found that the work has a few flaws and is slightly premature for publication. Firstly, it relies heavily on specific behavior of the ART dataset, specifically, multiple gold positives and multiple gold negative examples in the training set. Also, the experiments need to be improved a bit to better disentangle the impact of late-stage interaction and focal losses on the performance of this task. Finally, besides the very low resource setting, it's not clear that the proposed changes make a significant impact in the performance of task; the reported gains seem quite modest.  My recommendation to the authors is to potentially improve this work by focusing on the low-resource setting, where the proposed techniques show some potential improvements in sample efficiency (see comments above for specific recommendations on how one could explore such a study).

---

### Decision · Program_Chairs · 2022-01-20

**Decision:**

Reject

**Comment:**

The paper studies improving model for abductive natural language inference task. Specifically, they introduce information interaction layers and the joint softmax focal loss.

On positive notes, their method shows persuasive empirical gains. However, reviewers found (1) the technical novelty of the approach to be limited (reviewer croc, 3Vwo, W1Sp), (2) approaches (especially focal loss) not well motivated (reviewer hk5y), (3) there are limited take-away from the paper (reviewer imYG, hk5y) and (4) claims not well supported and experimental details missing (reviewer hk5y). The reviewers further provided detailed comments that would be helpful for authors to improve the paper. Because of such limitations, in its current form, the paper is not ready for publication.